# Observational study of factors associated with morbidity and mortality from COVID-19 in Lebanon, 2020–2021

Moni Nader[1], Omar Zmerli[2], Daniel E. Platt[3], Hamdan Hamdan[1], Salwa Hamdash[4], Rami Abi Tayeh[2], Jad Azar[2], Diana Kadi[2], Youssef Sultan[2], Taha Bazarbachi[4], Gilbert Karayakoupoglou[4], Pierre Zalloua[1,5]*, Eid Azar[2]*

**1** College of Medicine and Health Sciences, Khalifa University, Abu Dhabi, United Arab Emirates, **2** Division of Infectious Diseases, Department of Medicine, Saint George Hospital University Medical Center, Beirut, Lebanon, **3** Computational Biology Center, IBM TJ Watson Research Centre, Yorktown Hgts, New York, United States of America, **4** Laboratory Medicine, Haykel Hospital, Tripoli, Lebanon, **5** Harvard T.H. Chan School of Public Health, Boston, Mssachusets, United States of America

* eeazar@stgeorgehospital.org (EA); pierre.zalloua@ku.ac.ae (PZ)

## Abstract

### Background

The COVID-19 pandemic claimed millions of lives worldwide without clear signs of abating despite several mitigation efforts and vaccination campaigns. There have been tremendous interests in understanding the etiology of the disease particularly in what makes it severe and fatal in certain patients. Studies have shown that COVID-19 patients with kidney injury on admission were more likely to develop severe disease, and acute kidney disease was associated with high mortality in COVID-19 hospitalized patients.

### Methods

This study investigated 819 COVID-19 patients admitted between January 2020-April 2021 to the COVID-19 ward at a tertiary care center in Lebanon and evaluated their vital signs and biomarkers while probing for two main outcomes: intubation and fatality. Logistic and Cox regressions were performed to investigate the association between clinical and meta-bolic variables and disease outcomes, mainly intubation and mortality. Times were defined in terms of admission and discharge/fatality for COVID-19, with no other exclusions.

### Results

Regression analysis revealed that the following are independent risk factors for both intubation and fatality respectively: diabetes (p = 0.021 and p = 0.04), being overweight (p = 0.021 and p = 0.072), chronic kidney disease (p = 0.045 and p = 0.001), and gender (p = 0.016 and p = 0.114). Further, shortness of breath (p<0.001), age (p<0.001) and being overweight (p = 0.014) associated with intubation, while fatality with shortness of breath (p<0.001) in our group of patients. Elevated level of serum creatinine was the highest factor associated with

**Data Availability Statement:** All relevant data are within the article and its Supporting information files.

**Funding:** The author(s) received no specific funding for this work.

**Competing interests:** The authors have declared that no competing interests exist.

fatality (p = 0.002), while both white blood count (p<0.001) and serum glutamic-oxaloacetic transaminase levels (p<0.001) emerged as independent risk factors for intubation.

## Conclusions

Collectively our data show that high creatinine levels were significantly associated with fatality in our COVID-19 study patients, underscoring the importance of kidney function as a main modulator of SARS-CoV-2 morbidity and favor a careful and proactive management of patients with elevated creatinine levels on admission.

## Introduction

The newly manifested Severe Acute Respiratory Syndrome Coronavirus 2 (SARS-CoV-2) pandemic was initially associated with SARS (COVID-19). As the pandemic progressed, SARS-CoV-2 infections varied from asymptomatic, to mild flu-like, to severe symptoms leading to death in about 2% of cases (https://www.who.int/health-topics/coronavirus). In January 2020, The World Health Organization (WHO) declared the COVID-19 outbreak a public health emergency of international concern, and a global pandemic in March 2020. To date, there have been more than 600 million confirmed COVID-19 cases, with over six million deaths Worldwide (https://covid19.who.int). The management of the disease varied from patient to patient [1] mostly due to the paucity of the clinical information and the course of the disease manifestation at the onset of the pandemic, but more importantly from the lack of categorization of risk factors that were associated with high morbidities and mortality.

The viral infection triggered cytokine storm in COVID-19 patients, along with chronic conditions such as cancer, diabetes, lung disease, kidney disease, and cardiovascular disease were critical in determining COVID-19 outcomes. Other factors including obesity and gender have been shown to correlate with the severity of COVID-19 symptoms. The sickest COVID-19 patients, or those who were admitted to the intensive care unit (ICU), were often obese subjects irrespective of their age [2]. Several common factors associated with COVID-19 severity have been established across many populations, while some were shown to be population-specific [3, 4]. As an example, the male to female fatality ratio in many populations was estimated at 3.5 [5–7], except in India where COVID-19 fatality rate was shown to be higher in women than men (3.3% versus 2.9%) [8, 9].

The SARS-CoV-2 infection is not limited to the upper respiratory tract and lungs, but it commonly affects multiple organs including the heart, liver, and kidneys [10–12]. Multiorgan failure has been associated with severe COVID-19 cases worldwide. Hospitalized COVID-19 patients who often presented, upon admission, with acute respiratory distress syndrome disseminated intravascular coagulation, venous thromboembolism, bone marrow failure, acute kidney injury, myocarditis, neurological manifestations, among others [13]. Overall, a significant number of COVID-19 patients developed acute kidney injury (AKI) secondary to COVID-19. These variabilities in the clinical manifestations of COVID-19 call for a necessity to investigate various risk factors associated with fatality in COVID-19 patients and in different populations to account for inter-patient, clinical, and demographic variabilities.

According to the Ministry of Public Health in Lebanon, there have been more than 1.2 M confirmed COVID-19 cases and more than 10,000 COVID-19 associated deaths. Although more than 20% of the population was infected, and close to 2% of the confirmed infections were fatal, there was no systemic study that explored the risk factors associated with COVID-

19 in Lebanon. Herein, we evaluated the clinical manifestations in a group of COVID-19 patients with critical/fatal and non-critical illness from one of the largest tertiary care centers in Lebanon. We investigated and identified several risk factors for their association with COVID-19 severity and/or fatality. These factors can be used to build a comprehensive clinical platform for a better management of COVID-19 patients.

## Materials and methods

### Patients

We investigated 819 COVID-19 patients who were admitted between January 2020 and April 2021 to the COVID-19 ward at a major tertiary care hospital in Lebanon. This hospital was one of the major hospitals in the Lebanese capital that was dedicated to treating COVID-19 patients who came not only from the greater Beirut area but from across the entire country. Patients were evaluated upon admission and were admitted to the COVID-19 ward if they tested positive for the SARS-CoV-2 by quantitative real-time reverse transcriptase-polymerase chain reaction (qRT-PCR) of nasopharyngeal swabs using LightMix Modular SARS-CoV-2 E, N, and RdRP-genes (TibMolbiol, Berlin, Germany) performed according to manufacturer's manual. COVID-19 diagnosis was made according to the WHO guidance (https://www.who.int/publications/i/item/diagnostic-testing-for-sars-cov-2). Anonymized electronic medical records of all patients were retrieved and reviewed for epidemiological, demographic, clinical and laboratory data.

After weight determination and height measurements, the body mass index (BMI) was calculated, and patients were evaluated for the following clinical manifestations: Abdominal pain, diarrhea, anosmia, fever, chills, fatigue, myalgia, headaches, cough, sore throat, and shortness of breath. Patients' medical history were assessed for the following medical conditions: Asthma, chronic obstructive pulmonary disease (COPD), cancer, chronic kidney disease (CKD), cardiovascular disease (CVD), diabetes, hypertension, and autoimmune disease. Oxygen requirement, the need for and the duration of intubation, and the length of hospital stay until discharge or death were recorded.

White blood and platelets counts were obtained on hospital admission. Liver function tests (serum glutamic-oxaloacetic transaminase (SGOT) or aspartate aminotransferase (AST), and serum glutamic pyruvic transaminase (SGPT) or alanine aminotransferase (ALT)); high-sensitivity C-reactive protein (CRP); hemoglobin (Hg); Serum Creatinine kinase; D-dimer and ferritin levels were also determined on admission. The two main outcome variables tested were fatality or "severe" COVID-19 (defined as requiring intubation).

### Statistical analysis

Data were managed using pandas (version 1.1.2) [14]. Logistic regressions were performed using statsmodels (statsmodels.discrete.discrete_model.Logit version 0.12.0) [15] and Cox regressions were performed using coxph in R (survival version 3.2.1, survminer version 0.4.9) [16]. Times were defined in terms of admission and discharge/fatality for COVID-19, with no other exclusions. Only records that completed discharge or fatality were included in all analyses. There were no outpatient follow-ups.

Categorical and quantitative variable definitions are as follows: PlateletsC (150–400, Scaled and centered), VolumeC (Scaled and centered), CRPC (Normal <10, Scaled and centered), HbC (Normal 13.8–17.2 (men), 12.1–15.1 (women), Scaled and centered), AgeC (1 if age $\geq$65, else 0), WBCC (WBC $\geq$ 11), CreatinineC (Creatinine > 1 for women, > 1.2 for men), SGOTC (SGOT (AST) $\geq$43 for women, 48 for men), SGPTC (SGPT (ALT) $\geq$45 for women, 55 for men), SexC (Sex—1; Sex = 2 for women, 1 for men), DDimerC ($\geq$ 250), Ferritin ($\geq$ 160

women, ≥300 for men), Obesity (>30) (S1 Table). Summaries of the quantitative/categorical data were generated using pandas.

## Ethical considerations

This study was approved by the Ethics Committee of University of Balamand/Saint George Hospital University Medical Center (IRB-REC/O1018-2011320). Consenting to participate in the study was not required, and all patient data were anonymized before it was transferred to the research team for analyses. This study was granted an exemption from requiring informed consent by the Ethics Committee of University of Balamand/Saint George Hospital University Medical Center (IRB-REC/O1018-2011320).

## Results

In total, 819 hospital-admitted COVID-19 patients were enrolled in this study with 546 male (M) and 273 female (F) patients (66.77% M and 33.33% F). The comparison for each of the variables presented in Tables 1 and 2 showed no difference between the male and female groups, with admission female to male ratio being 1/3, suggesting that gender did not influence these variables in our population. Table 2 shows that there were more intubated males with shortness of breath than female (23.18 versus 16.42%, and 72.48 versus 59.78%, respectively). Males who presented with anosmia were almost double than females (3.30% versus 1.48%), albeit anosmia constituted a very small percentage of the hospitalized patients. Female patients were four times more likely to have autoimmune disease than males (8.21% versus 2.94%).

The fatality rate between male and female COVID-19 patients however was not significant (27% and 23% respectively, Fig 1).

### Regression analyses

**Common comorbidities for intubation and fatality.** Overall, the intubation of COVID-19 subjects was highly associated with fatality (OR 71.20, 95%CI[41.33–122.64], p <0.001) and with the oxygen consumption volume (OR, 3.18 95%CI[0.02–0.06], p<0.001) (Table 3).

We interrogated hypertension, diabetes, being overweight (BMI $\geq$ 25kg/m$^2$), chronic kidney disease (CKD), CVD, autoimmune disease, sex, asthma, and age against the prevalence of

**Table 1. Descriptive statistics of the continuous variables in the COVID-19 population.**

|  | Mean | Std Dev | F Mean | F Std Dev | M Mean | M StdDev |
|---|---|---|---|---|---|---|
| **Age (years)** | 61.6 | 17.8 | 61.5 | 20.5 | 61.6 | 16.4 |
| **Weight (kg)** | 83.2 | 20.7 | 74.6 | 19.9 | 87.3 | 19.8 |
| **Height (m)** | 1.6 | 0.1 | 1.6 | 0.1 | 1.7 | 0.1 |
| **BMI (kg/m^2)** | 28.7 | 5.4 | 28.5 | 6.2 | 28.8 | 5.1 |
| **WBC (10^3/μl)** | 9.3 | 5.2 | 9.1 | 5.9 | 9.4 | 4.7 |
| **Hb (g/dL)** | 13.1 | 2.1 | 12.0 | 1.8 | 13.6 | 2.1 |
| **Platelets(10^3/μl)** | 238.0 | 106.5 | 247.4 | 118.9 | 233.4 | 99.5 |
| **D Dimer (ng/mL)** | 1.3 | 3.3 | 1.2 | 1.7 | 1.3 | 3.9 |
| **Creatinine(mg/dl)** | 1.2 | 1.0 | 1.1 | 1.1 | 1.2 | 1.0 |
| **SGOT (AST)** | 66.6 | 212.2 | 72.0 | 269.9 | 64.0 | 179.4 |
| **SGPT (ALT)** | 54.1 | 131.6 | 49.8 | 132.1 | 56.0 | 131.5 |
| **Ferritin (ng/ml)** | 906.9 | 661.6 | 657.2 | 634.1 | 1029.9 | 640.6 |
| **CRP (mg/l)** | 7.0 | 8.3 | 5.9 | 8.6 | 7.5 | 8.1 |

**Table 2. Descriptive statistics of the COVID-19 population stratified by medical condition.**

| | Count | Rel Freq % | F Count | F Rel Freq % | M Count | M Rel Freq % |
|---|---|---|---|---|---|---|
| **Total** | 819 | 100.00 | 273.00 | 100.00 | 546.00 | 100.00 |
| **SexC** | 273 | 33.33 | 273.00 | 100.00 | 0.00 | 0.00 |
| **AgeC** | 390 | 47.62 | 136.00 | 49.82 | 254.00 | 46.52 |
| **FatalityC** | 213 | 26.01 | 64.00 | 23.44 | 149.00 | 27.29 |
| **Intubated** | 168 | 20.92 | 44.00 | 16.42 | 124.00 | 23.18 |
| **Asthma** | 29 | 3.55 | 12.00 | 4.43 | 17.00 | 3.11 |
| **COPD** | 37 | 4.53 | 11.00 | 4.06 | 26.00 | 4.76 |
| **Cancer** | 70 | 8.57 | 33.00 | 12.18 | 37.00 | 6.78 |
| **Chronic kidney disease** | 88 | 10.77 | 29.00 | 10.70 | 59.00 | 10.81 |
| **CVD** | 282 | 34.52 | 78.00 | 28.78 | 204.00 | 37.36 |
| **Diabetes** | 232 | 28.61 | 78.00 | 29.10 | 154.00 | 28.36 |
| **Hypertension** | 418 | 51.41 | 130.00 | 48.51 | 288.00 | 52.84 |
| **Autoimmune disease** | 38 | 4.67 | 22.00 | 8.21 | 16.00 | 2.94 |
| **Overweight** | 471 | 57.51 | 142.00 | 52.01 | 329.00 | 60.26 |
| **Obese** | 216 | 26.37 | 72.00 | 26.37 | 144.00 | 26.37 |
| **Abdominal pain/ diarrhea** | 155 | 19.00 | 59.00 | 21.77 | 96.00 | 17.61 |
| **Anosmia** | 22 | 2.70 | 4.00 | 1.48 | 18.00 | 3.30 |
| **fever/ Chills** | 512 | 62.82 | 149.00 | 54.98 | 363.00 | 66.73 |
| **Fatigue** | 261 | 32.18 | 78.00 | 28.89 | 183.00 | 33.83 |
| **Myalgia** | 268 | 32.88 | 79.00 | 29.15 | 189.00 | 34.74 |
| **Headaches** | 72 | 8.83 | 16.00 | 5.90 | 56.00 | 10.29 |
| **Cough** | 413 | 50.67 | 127.00 | 46.86 | 286.00 | 52.57 |
| **Sore throat** | 50 | 6.13 | 11.00 | 4.06 | 39.00 | 7.17 |
| **Shortness of breath** | 557 | 68.26 | 162.00 | 59.78 | 395.00 | 72.48 |

F: female, M: male, Rel Freq: Relative Frequency.

intubation in COVID-19 patients. We found that diabetes (OR 1.57, 95%CI[1.07–2.32], p = 0.02), being overweight (OR 0.65, 95%CI[0.46–0.94], p = 0.02), CKD (OR 1.70, 95%CI [1.01–2.84], p = 0.04), and sex (OR 0.61, 95%CI[0.41–0.91], p = 0.016) were independent risk factors for intubation, with age conferring the most significant intubation risk (OR 2.03, 95% CI[1.34–3.08], p = 0.001) (Table 4). When assessing for fatality as an outcome variable (Table 5), a similar trend was noticed, with age ($\geq$ 65 years) scoring the highest significance as an independent risk factor (OR 2.93, 95%CI[1.97–4.34], p<0.001). However, being overweight did not show significance (p = 0.07) in this analysis.

**Symptoms at presentation with intubation and fatality.** We further evaluated abdominal pain/diarrhea, fever/chills, fatigue, myalgia, headaches, cough, rhinorrhea, sore throat, and shortness of breath in our group of COVID-19 patients. Having shortness of breath (OR 2.97, 95%CI[1.87–4.70], p<0.001), age (OR 2.60, 95%CI[1.78–3.80], p<0.001), and gender (OR 0.64, 95%CI[0.43–0.96], p = 0.03) was significantly associated with intubation of these patients (Tables 6 and 7). Shortness of breath on admission was one of the most important risk factors for patients requiring intubation. Furthermore, mortality from COVID-19 was significantly associated with shortness of breath (OR 2.63, 95%CI [1.76–3.93], p<0.001).

**Biomarkers of intubation and COVID-19 fatality.** In evaluating patient biomarkers related to need for intubation, results showed that CRP (OR 1.48, 95%CI [1.21–1.81], p<0.001), WBC (OR 2.70, 95%CI [1.68–4.32], p<0.001), SGOT (OR 3.08, 95%CI [1.88–5.03],

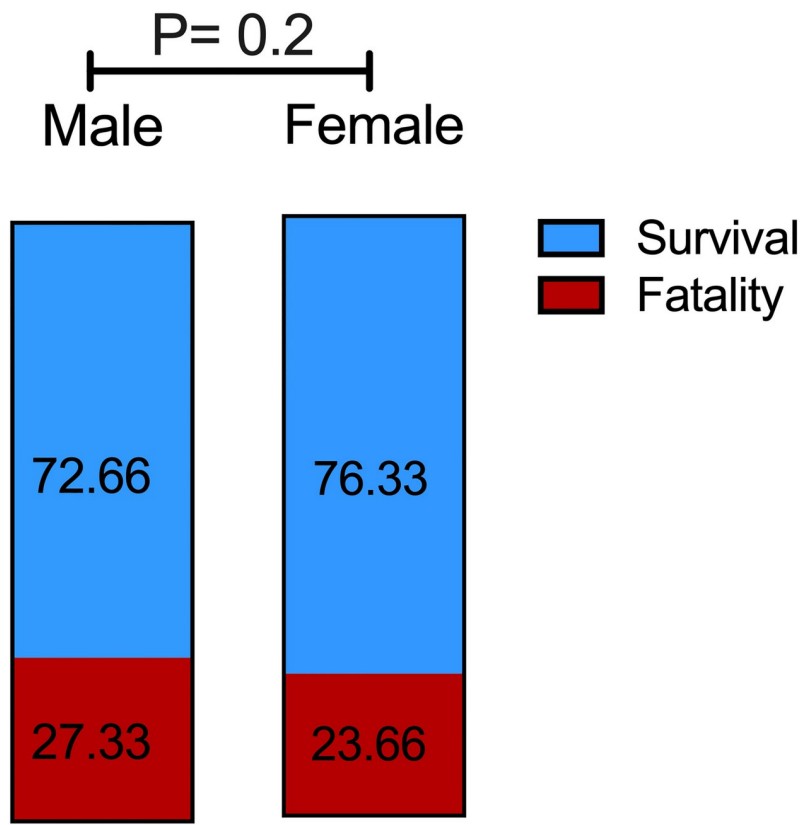

**Fig 1. Histograms showing the percentage of fatality in male and female admitted to the ICU.** P <0.02 was determined using the Chi-Square test.

p<0.001), and SGPT (ALT) (OR 0.44, 95%CI [0.23–0.81], p = 0.01) were independent biomarkers for the intubation of these patients, while platelets, ferritin, Hb, and creatinine did not score significance. Of these biomarkers WBC and SGOT (AST) were the most significantly associated with intubation (Table 8). Bivariate analysis of the same biomarkers found platelets (OR 0.78, 95%CI [0.61–0.97], p = 0.029), CRP (OR 1.39, 95%CI [1.14–1.69], p = 0.001), creatinine (OR 2.02, 95%CI [1.30–3.12], p = 0.002), WBC (OR 3.25, 95%CI [2.07–5.08], p<0.001), and SGOT (AST) (OR 2.18, 95%CI [1.35–3.5], p = 0.001) were associated with mortality in COVID-19 patients. WBC, platelets, and creatinine levels were among the most significant biomarkers associated with mortality (Table 9).

Creatinine levels and/or diabetes were strongly associated with hypertension in our group of patients. Similarly, diabetes and hypertension were associated with high creatinine levels, while hypertension was strongly associated with kidney disease (Table 10).

**Table 3. Binomial logistic regression odds ratios predicting fatality from oxygen volume requirement and intubation.**

| | O₂ Volume | | | | Intubation | | | |
|---|---|---|---|---|---|---|---|---|
| | **OR** | **95CI-** | **95CI+** | **P-val** | **OR** | **95CI-** | **95CI+** | **P-val** |
| **Fatality** | - | - | - | - | 71.201 | 41.337 | 122.64 | p<0.001 |
| **Fatality** | 3.185 | 2.393 | 4.240 | p<0.001 | - | - | - | - |

**Table 4. Binomial logistic regression odds ratios of discrete variables predicting intubation from joint conditions as exposures.**

| Severity | OR | 95CI- | 95CI+ | P-val |
|---|---|---|---|---|
| Hypertension | 1.049 | 0.703 | 1.565 | 0.814 |
| Diabetes | 1.577 | 1.071 | 2.322 | 0.021 |
| Overweight | 0.653 | 0.455 | 0.937 | 0.021 |
| Asthma | 0.495 | 0.143 | 1.719 | 0.269 |
| Chronic kidney disease | 1.697 | 1.013 | 2.844 | 0.045 |
| CVD | 1.103 | 0.731 | 1.664 | 0.640 |
| Autoimmune disease | 0.555 | 0.189 | 1.630 | 0.284 |
| SexC | 0.609 | 0.407 | 0.911 | 0.016 |
| AgeC | 2.029 | 1.338 | 3.079 | 0.001 |

**Table 5. Binomial logistic regression coefficients of discrete variables predicting fatality from joint conditions as exposures.**

| Fatality | OR | 95CI- | 95CI+ | P-val |
|---|---|---|---|---|
| Hypertension | 1.111 | 0.762 | 1.619 | 0.586 |
| Diabetes | 1.471 | 1.019 | 2.124 | 0.040 |
| Overweight | 0.730 | 0.518 | 1.028 | 0.072 |
| Asthma | 0.338 | 0.097 | 1.182 | 0.089 |
| Chronic kidney disease | 2.253 | 1.378 | 3.684 | 0.001 |
| CVD | 1.373 | 0.938 | 2.008 | 0.103 |
| Autoimmune disease | 1.652 | 0.779 | 3.507 | 0.191 |
| SexC | 0.740 | 0.510 | 1.074 | 0.114 |
| AgeC | 2.927 | 1.972 | 4.344 | <0.001 |

**Table 6. Binomial logistic regression coefficients of various factors predicting intubation from joint symptoms as exposures.**

| Severity | OR | 95CI- | 95CI+ | P-val |
|---|---|---|---|---|
| Abdominal pain/ diarrhea | 1.324 | 0.838 | 2.091 | 0.229 |
| fever/ Chills | 0.753 | 0.511 | 1.111 | 0.153 |
| Overweight | 0.632 | 0.437 | 0.913 | 0.014 |
| Fatigue | 0.850 | 0.556 | 1.299 | 0.452 |
| Myalgia | 1.000 | 0.647 | 1.544 | 0.998 |
| Headaches | 0.490 | 0.205 | 1.172 | 0.109 |
| Cough | 1.110 | 0.764 | 1.613 | 0.583 |
| Rhinorrhea | 1.895 | 0.784 | 4.581 | 0.156 |
| Sore throat | 1.206 | 0.534 | 2.723 | 0.653 |
| Shortness of breath | 2.975 | 1.878 | 4.712 | <0.001 |
| AgeC | 2.608 | 1.786 | 3.807 | <0.001 |
| SexC | 0.637 | 0.424 | 0.957 | 0.030 |

Finally, applying a Cox regression of time to death on the time-constant covariates (age, WBC, asthma and overweight) revealed a significant implication of these factors as shown in the predictive survival proportion, at any given point, for the above-mentioned covariates (Fig 2).

**Table 7. Binomial logistic regression coefficients of various factors predicting fatality from joint symptoms as exposures.**

| Fatality | OR | 95CI- | 95CI+ | P-val |
|---|---|---|---|---|
| **Abdominal pain/ diarrhea** | 1.255 | 0.814 | 1.933 | 0.304 |
| **fever/ Chills** | 0.888 | 0.615 | 1.281 | 0.526 |
| **Overweight** | 0.713 | 0.506 | 1.006 | 0.054 |
| **Fatigue** | 0.870 | 0.588 | 1.287 | 0.485 |
| **Myalgia** | 1.151 | 0.770 | 1.721 | 0.493 |
| **Headaches** | 0.523 | 0.236 | 1.159 | 0.110 |
| **Cough** | 0.928 | 0.654 | 1.317 | 0.677 |
| **Rhinorrhea** | 1.504 | 0.623 | 3.630 | 0.364 |
| **Sore throat** | 0.885 | 0.392 | 1.998 | 0.769 |
| **Shortness of breath** | 2.628 | 1.756 | 3.932 | <0.001 |
| **AgeC** | 4.239 | 2.957 | 6.075 | <0.001 |
| **SexC** | 0.779 | 0.537 | 1.129 | 0.187 |

**Table 8. Binomial logistic regression coefficients of continuous variables predicting intubation from joint blood panels as exposures.**

| Severity | OR | 95CI- | 95CI+ | P-val |
|---|---|---|---|---|
| **PlateletsC** | 0.927 | 0.738 | 1.166 | 0.519 |
| **FerritinC** | 2.102 | 0.907 | 4.872 | 0.083 |
| **CRPC** | 1.484 | 1.214 | 1.813 | <0.001 |
| **HbC** | 0.915 | 0.725 | 1.156 | 0.457 |
| **CreatinineC** | 1.274 | 0.790 | 2.055 | 0.321 |
| **WBCC** | 2.701 | 1.689 | 4.320 | <0.001 |
| **SGOTC** | 3.077 | 1.880 | 5.038 | <0.001 |
| **SGPTC** | 0.440 | 0.236 | 0.819 | 0.010 |

**Table 9. Binomial logistic regression coefficients of continuous variables predicting fatality from joint blood panels as exposures.**

| Fatality | OR | 95CI- | 95CI+ | P-val |
|---|---|---|---|---|
| **PlateletsC** | 0.776 | 0.618 | 0.974 | 0.029 |
| **FerritinC** | 1.210 | 0.635 | 2.305 | 0.562 |
| **CRPC** | 1.390 | 1.142 | 1.691 | 0.001 |
| **HbC** | 0.833 | 0.672 | 1.032 | 0.095 |
| **CreatinineC** | 2.021 | 1.308 | 3.125 | 0.002 |
| **WBCC** | 3.246 | 2.073 | 5.081 | <0.001 |
| **SGOTC** | 2.178 | 1.355 | 3.501 | 0.001 |
| **SGPTC** | 0.588 | 0.325 | 1.062 | 0.078 |

## Discussion

The SARS-CoV-2 pandemic and its resulting COVID-19 illness have affected the lives of millions around the globe posing a severe worldwide public health challenge [1]. Patients with underlying comorbidities were at higher risk, particularly those with cardiac conditions, diabetes, and kidney injury that necessitated a pressing need to mitigate viral spread [10, 11]. While treatment protocols are now starting to converge, the identification of the numerous factors

**Table 10. Interaction of the different risk factors of COVID-19 probed against fatality.**

| | Diabetes | | | | Hypertension | | | | Diabetes & Hypertension | | | | Creatinine | | | |
|---|---|---|---|---|---|---|---|---|---|---|---|---|---|---|---|---|
| | OR | 95CI- | 95CI+ | P-val | OR | 95CI- | 95CI+ | P-val | OR | 95CI- | 95CI+ | P-val | OR | 95CI- | 95CI+ | P-val |
| Chronic kidney disease | - | - | - | - | - | - | - | - | - | - | - | - | 14.034 | 7.956 | 24.75 | <0.001 |
| Chronic kidney disease | 0.981 | 0.593 | 1.621 | 0.09 | 1.636 | 1.017 | 2.630 | 0.04 | - | - | - | - | - | - | - | - |
| Chronic kidney disease | 0.994 | 0.368 | 2.684 | 0.98 | 1.642 | 0.960 | 2.808 | 0.07 | 0.983 | 0.311 | 3.108 | 0.97 | - | - | - | - |
| Creatinine | 1.644 | 1.173 | 2.305 | <0.001 | 2.887 | 2.070 | 4.024 | <0.001 | - | - | - | - | - | - | - | - |
| Creatinine | 1.448 | 0.747 | 2.807 | 0.27 | 2.763 | 1.881 | 4.057 | <0.001 | 1.189 | 0.550 | 2.572 | 0.65 | - | - | - | - |
| Hypertension | 3.908 | 2.790 | 5.473 | 0.002 | - | - | - | - | - | - | - | - | - | - | - | - |
| Hypertension | - | - | - | - | - | - | - | - | - | - | - | - | 3.263 | 2.367 | 4.497 | <0.001 |
| Hypertension | 3.498 | 2.476 | 4.941 | <0.001 | - | - | - | - | - | - | - | - | 2.887 | 2.071 | 4.024 | <0.001 |

leading to COVID-19 morbidity and mortality remains under extensive investigation. In our study group, kidney disease, diabetes, and age were independently associated with fatality while, platelets, CRP, WBC, SGOT and high levels of serum creatinine biomarker of kidney injury, were also associated with fatality among COVID-19 hospitalized hypertensive patients.

It has been established that males are more prone to hospitalization, intubation, and death than women [17]. While we noticed a hospitalization ratio of 3:1 in males versus female in our study population, the death rate post hospitalization between both sexes was similar.

The rate of autoimmune disease in the hospitalized COVID-19 population was substantially higher than those observed for common autoimmune diseases around the world (https://nationalstemcellfoundation.org/glossary/autoimmune-disease/). While autoimmune diseases tend to be more prevalent in women more than men, being female tended to be protective in our dataset [18, 19]. Our results suggest that autoimmune diseases may have promoted hospitalization, but the small numbers of patients with autoimmune diseases limited our ability to test significance.

Our patients showed increased WBC was significantly associated with both intubation and fatality of these patients. The levels of WBC in the settings of COVID-19 hospitalization remain debatable and the interaction between the immune system and SARS-CoV-2 is not yet fully understood. While studies report a decrease in WBC, others showed an association between WBC and mortality in COVID-19 patients [20–24].

We also note that ALT was significantly elevated among severe COVID-19 cases, but AST was not. This pattern is typical for ischemic hepatic damage or skeletal/cardiac muscle [25] breakdown in contrast with toxicity induced damage [26], and would be expected among patients with severe COVID-19. This ALT predominance further highlights the extrahepatic impact caused by this virus, possibly causing myositis. Our data is in accord with other studies showing that ALT levels are increased in severe cases of COVID-19 [27], and that high levels of ALT correlate with ICU admission in COVID-19 patients [28, 29].

Our finding that elevated creatinine, predominately in hypertensive patients, was highly associated with intubation and fatality of COVID-19 patients is of great importance since hypertension is rampant among adults in Lebanon [30]. This is in accord with other reports showing a 2.5-fold increased risk of severity and mortality in hypertensive COVID-19 [31].

The association between hypertension, elevated creatinine and COVID-19 severity may be related to underlying kidney disease, hypovolemic kidney injury and or some failure of the renin-angiotensin system in response to SARS-CoV-2 [32]. Yet, acute increase in blood pressure is also manifested in COVID-19 subjects [33] and whether hypertension is a risk factor for fatality of these patients, or whether it is a vital parameter indicative of severity remains to be determined.

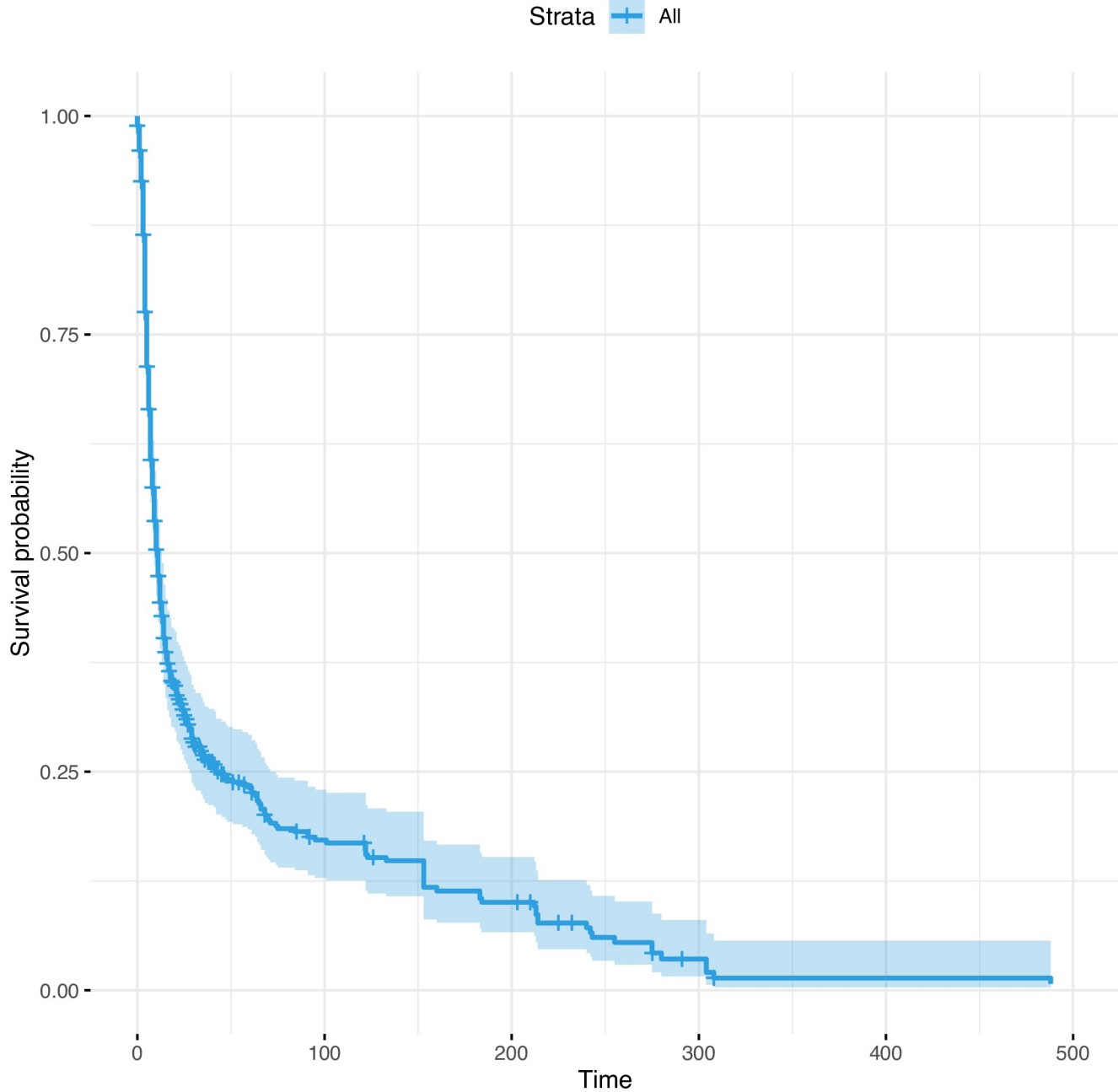

**Fig 2. Cox regression predicting the hazard ratio from overweight, asthma, and high white blood count.**

These results, however, point to a synergistic effect between kidney injury and hypertension that is modulating the progression of the COVID-19 illness. In fact, there was a strong link between increased levels of creatinine and kidney disease in our study group. A relationship that is in line with common medical practices adopting increased creatinine levels, chief among other factors as biomarker for kidney injury and the major component of eGFR [34]. Elevated levels of creatinine scored the highest odds among all other risk factors in our study

group, suggesting that kidney function upon admission may be an important prognostic determinant of COVID-19 disease.

Our findings are in accord with others showing that acute kidney injury (AKI) is associated with high mortality rate in COVID-19 subjects [35]. AKI is a clinical syndrome manifested by a decline in kidney function and is closely associated with morbidity and mortality [36]. AKI was observed in 5%–15% of SARS and MERS cases with a high mortality rate (60%–90%). Some reports indicated that the incidence of AKI is inconsistent among patients with COVID-19 [37, 38]. It was recently reported that COVID-19 patients with high levels of serum creatinine were more susceptible to develop AKI during hospitalization, an observation coherent with similar findings in SARS subjects [35, 39]. The SARS-CoV-2 particles invade the kidneys and induce cellular damage leading to renal dysfunction [10]. These viral particles were found in kidney tissues (autopsies) associated with podocytes injury leading to acute tubular necrosis, protein leakage in Bowman's capsule, collapsing glomerulopathy, and mitochondrial impairment [40]. SARS-CoV-2 can also infiltrate the renal tissues along with inflammatory cells expressing CD147, thus leading to dysregulation of the cell cycle and inflammatory response [41]. It remains undiscernible whether AKI was caused by the SARS-CoV-2/cytokine storm, or whether our patients had CKD prior to their SARS-CoV-2 infection which exacerbated their condition. Autopsies were not performed on Lebanese COVID-19 patients who died from the disease, and therefore the presence of SARS-CoV-2 in the kidneys could not be documented. It is however evident that kidney function must be closely monitored in COVID-19 patients. The strong association of increased creatinine levels and fatality in COVID-19 patients poses a high risk to subjects on hemodialysis who could potentially contract the SARS-CoV-2.

When measuring the interaction between hypertension, T2D and high creatinine levels, each of the variables (hypertension or T2D) highly associated with increased creatinine levels when probing for intubation and fatality. Our data shows that creatinine levels and/or diabetes are strongly linked to hypertension, but diabetes and hypertension together did not show a significant association neither with kidney disease nor with high creatinine levels. Therefore, elevated creatinine levels, hence, kidney injury, remain associated with fatality namely in hypertensive subjects. T2D did not promote the infection by SARS-CoV-2 however it exacerbates the condition in COVID-19 patients [42]. When investigating COVID-19 severity risk with hypertension and T2D in an interactive analysis, the increased creatinine levels failed to significantly associate with intubation and fatality. Thus, our data suggest that diabetic patients who are not hypertensive and with elevated creatinine levels were less likely to get intubated or die from COVID-19 compared to hypertensive patients with elevated creatinine levels. Hypertension and obesity were also significantly associated with the severity of the disease and with high mortality rate, possibly due to their higher frequency in the Lebanese population.

It is worth noting that all COVID-19 patients in this study were not vaccinated. Although, the disparities in the spread and infectivity of the different SARS-CoV-2 variants are highly related to mutations/modifications in the viral proteins involved in the interaction with the host cell [43], it remains unclear what host factors modulate the course of infection.

The absence of creatinine baseline levels prior to admission was a major limitation of the study. Moreover, markers of kidney function (other than creatinine) were not assessed upon admission. This was partly mitigated by the information collected on CKD upon admission. The disease evolution could not be assessed against creatinine levels as additional data points on creatinine levels during hospitalization were not available. In addition, follow up data on patients after discharge was not available. Additional studies in multiple populations are needed to demonstrate whether high creatinine levels constitute a generalizable risk for mortality in hypertensive COVID-19 patients.

In conclusion, we found that hypertension and increased creatinine levels are comorbid factors that cannot be ignored in COVID-19 subjects in view of their significant association with intubation and mortality. We therefore advocate for a complete appraisal of the biomarkers underlining kidney injury and hypertension to guide a therapeutic strategy for COVID-19. We also recommend a firm monitoring of markers of kidney injury upon COVID-19 patients' admission to avoid major complications. Finally, special attention must be given to hypertensive subjects with reduced kidney function as these may be some of the most vulnerable group. Since the number of fully vaccinated subjects in Lebanon has not yet reached the 50% mark, it is therefore imperative to actively target patients with acute or chronic kidney injury for a full vaccination regimen to avoid COVID-19 associated morbidities and mortalities.

## Supporting information

**S1 Table. Categorical and quantitative variable definitions.**
(DOCX)

**S1 Data.**
(XLSX)

## Author Contributions

**Conceptualization:** Pierre Zalloua, Eid Azar.

**Data curation:** Moni Nader, Omar Zmerli, Daniel E. Platt, Hamdan Hamdan, Salwa Hamdash, Rami Abi Tayeh, Jad Azar, Diana Kadi, Youssef Sultan, Taha Bazarbachi, Gilbert Karayakoupoglou, Pierre Zalloua, Eid Azar.

**Formal analysis:** Moni Nader, Daniel E. Platt, Salwa Hamdash, Pierre Zalloua, Eid Azar.

**Investigation:** Moni Nader, Pierre Zalloua.

**Methodology:** Moni Nader, Pierre Zalloua, Eid Azar.

**Supervision:** Pierre Zalloua, Eid Azar.

**Writing – original draft:** Moni Nader, Omar Zmerli, Hamdan Hamdan, Pierre Zalloua, Eid Azar.

**Writing – review & editing:** Moni Nader, Pierre Zalloua, Eid Azar.

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
