## [Decision Letter · Decision Letter 0]

24 Jun 2022

PONE-D-22-11784Creatinine levels on admission are main modulators of COVID-19 severityPLOS ONE

Dear Dr. Zalloua,

Thank you for submitting your manuscript to PLOS ONE. After careful consideration, we feel that it has merit but does not fully meet PLOS ONE’s publication criteria as it currently stands. Therefore, we invite you to submit a revised version of the manuscript that addresses the points raised during the review process.

Please see attached comments. The paper as currently written does not meet PLOS publication standards. The methods need further technical detail, the conclusions are not fully appropriate given the results, and the writing needs significant improvement.==============================

We look forward to receiving your revised manuscript.

Kind regards,

Eili Y. Klein, PhD

Academic Editor

PLOS ONE

Journal Requirements:

3. PLOS requires an ORCID iD for the corresponding author in Editorial Manager on papers submitted after December 6th, 2016. Please ensure that you have an ORCID iD and that it is validated in Editorial Manager. To do this, go to ‘Update my Information’ (in the upper left-hand corner of the main menu), and click on the Fetch/Validate link next to the ORCID field. This will take you to the ORCID site and allow you to create a new iD or authenticate a pre-existing iD in Editorial Manager. Please see the following video for instructions on linking an ORCID iD to your Editorial Manager account: https://www.youtube.com/watch?v=_xcclfuvtxQ.

5. Please include your tables as part of your main manuscript and remove the individual files. Please note that supplementary tables should remain as separate "supporting information" files.

Additional Editor Comments:

The paper and results are interesting, but the paper suffers from many severe flaws. Please change the title. This is a correlational analysis of ~800 patients not an RCT, thus the claim in the title is not correct and misleading. The writing throughout is poor and needs revision, please have it proofread more carefully. Additionally, the structure of the paper should adhere to scientific writing standards, do not put methods in the results, for example, clearly explain your cohort, etc, and put the tables in the paper not as some additional download.

On the methods, there needs to be significantly more included on the data, including, for example, what symptoms were extracted and when during their stay. The statistical analysis is also lacking in detail. Please refer to other papers published for advice on how to expand this section, but for example, "Multivariate logistic regressions were performed using statsmodels (ver 0.12.0) (17) and coxph in R (survival ver 3.2.1, survminer ver 0.4.9) (18)" tells the reader almost nothing about the methods and sticks in that a cox was performed as if it were a logistic regression. Also, why would data be managed in pandas but stats run in R, that makes little sense, statsmodels has a survival function. On the results and interpretation, just because something is correlated does not make it causative. Please remove any reference to creatinine levels being the causitive. The reviewers both have additional substantial concerns about the methods employed and you must add a robust limitations section.

Overall, the results are interesting, but the paper needs a tremendous amount of work for further consideration in this journal.

Reviewers' comments:

Reviewer's Responses to Questions

**Comments to the Author**

1. Is the manuscript technically sound, and do the data support the conclusions?

Reviewer #1: Partly

Reviewer #2: Partly

2. Has the statistical analysis been performed appropriately and rigorously? 

Reviewer #1: No

Reviewer #2: Yes

3. Have the authors made all data underlying the findings in their manuscript fully available?

Reviewer #1: No

Reviewer #2: Yes

4. Is the manuscript presented in an intelligible fashion and written in standard English?

Reviewer #1: No

Reviewer #2: No

5. Review Comments to the Author

Reviewer #1: The present paper explores different patient-level factors related to COVID-19 severity and fatality in a tertiary care centre in Lebanon.

Major comments

The abstract should be re-structured, accounting for the importance of creatinine on COVID-19 severity in the introduction, for example. Methods should reflect the statistical techniques used throughout the paper ( i.e., regression, correlation tests, etc.). The results lack statistical measures, including coefficients or p-values for the main predictors. The discussion should account for the step forward linking creatinine and COVID-19 severity and how policies could be oriented for better diagnosis or treatments, etc. Also, those correlations exhibited are positive? Negative? Etc. It’s worth expanding on the main results of the research in the abstract.

Could you please clarify the epidemiological, demographic, clinical and laboratory data retrieved in the methods section? This should be in order and right after you mention them all. Now you only mention clinical data. Also, the measurement might differ from study to study; would you mind adding more information on those characteristics and how they were measured? (It could be added in the supplementary material).

Could you please clarify the main outcomes and how they are measured in the methods section? The statistical analysis section is only for statistical inference approaches, and therefore, the “Categorical and quantitative variable definitions are defined in Table S1. The two main outcome variables tested were fatality or “severe” COVID-19…..” should be moved up to patients characteristics or data collection.

Statistical analyses. You speak about correlation in the abstract, but you used a logistic regression model, as this section depicts. Try to be clear on the regression method used. Did you test your outcomes? (i.e., their distributions, model residuals, multicollinearity, etc.). Also, did you use robust standard errors and if not, why?

Logistic model using severe COVID-19 as the outcome variable. Did you compare severe COVID-19 against those patients having mild COVID-19, asymptomatic COVID-19 and those who did not have it? I think this has to be clarified, and comparison groups should be ordered more clearly.

You applied a Cox regression for the mortality outcome. Could you please mention it in the statistical analysis subsection? Also, you might explain a bit about the primary predictors/exposures, model structure, how time was handled, censoring, etc.

The results section should be restructured. The explanation of variables incorporated in the modelling should be added in the methods section to facilitate readers’ understanding and information flow.

Supplementary material Table S1 should contain more detailed information. Were the variables analysed continuous or categorical? Why were the variables re-scaled, and does that impact the analyses? If they were re-scaled and centred, how and why did you do that?

No limitations are mentioned in the discussion section, and attention should be drawn to population characteristics and sample size, lack of weighting techniques or representability of the whole population, etc.

What are the study's main implications, or how can we connect this information with the current situation in Lebanon where people are being vaccinated (40% at least with one dose, 35% with two doses, and 9% with booster)?

Main results tables were not attached in the PDF file.

Minor comments

Please add the line numbers for the next revisions and to facilitate the review

Please consider revising the main text; there are some spelling, format and text errors. For instance, “..In January 2020, The..” : the “the” shouldn’t be written in capital letters.

Spell out WHO once first mentioned

The term ‘vs’ should be written in formal English.

Avoid the following words: “at the time of this writing”. Try to use the exact date to refer to the time and only if it is strictly needed.

Would you mind referring to the hospital analysed on the attended population, whether it is representative of the whole Lebanese population, etc.

If you first use WHO, then do not use its full name throughout the text.

Do not repeat over and over that 819 patients were admitted because it is mentioned in the methods and should be shown in descriptive stats tables.

Please spell out the M and F in the passage “66.77% M and 33.33% F”.

There are some inconsistencies with the use of decimal figures.

For p=0.000 is better to use p<0.001

There is no need to repeat ethical considerations two times throughout the manuscript.

Spell out the acronyms once first mentioned

Define that COVID-19 intensity is defined by whether patients were intubated.

Consider reframing the title by adding that it is based in a tertiary-care hospital in Lebanon.

Consider proofreading your manuscript.

Reviewer #2: Dear Authors

The current case report is interesting. But I have some opinions and doubts.

Title

Creatinine levels on admission are main modulators of COVID-19 severity.

In my opinion, it would be better to inform there is an association between severe COVID-19 and serum creatinine levels or kidney function.

In abstract

Results

Correlation analysis of various comorbidities revealed that hypertension, diabetes,

being overweight, kidney disease,.../What kind of kidney disease? Would it be CKD? AKI? Nephritis? Or AKD?

Methods

This study investigated 819 COVID-19 patients admitted to the COVID-19 ward at a

tertiary care center in Lebanon and evaluated their vital signs and biomarkers./ Are the biomarkers from hospital admission?

Materials and methods

Patients

Clinical data on admission included clinical symptoms, biochemical markers on admission, previous medical conditions.../

You should report what symptoms were studied, and what kinds of biochemical markers were used to compute.

What were the medical conditions you registered?

And about outcomes, did you compute the frequency of COVID-19-related AKI? And the need for kidney replacement therapy?

Results

In my opinion, you should put the results of the multivariate analysis in a table

-Common comorbidities for intubation and fatality

What was the type of kidney disease? Was it CKD? or glomerulonephritis with therapy immunosuppression?

We interrogated hypertension, diabetes, being overweight (BMI ≥ 25kg/m2), kidney disease, CVD...

What was the CI you used? Was it 95%?

Overall, the intubation of COVID-19 subjects was highly associated with fatality (OR 71.20, CI[41.33-122.64], p <0.001) and with the oxygen consumption volume (OR, 3.18 CI[0.02-0.06], p<0.001)… so forth

-Biomarkers of intubation and COVID-19 fatality

Again, what was the CI you used? Was it 95%?

We investigated serum levels of platelets, ferritin, CRP, Hb, creatinine, WBC, SGOT and SGPT and analyzed these using intubation as an independent variable. CRP (OR 1.48, CI [1.21-1.81], p=0.000), WBC (OR 2.70, CI [1.68-4.32], p=0.000), SGOT (OR 3.08, CI [1.88-5.03], p=0.000)./

I suggest to write p<0.001 rather than p=0.000. …so forth.

Discussion

In my opinion, you should report on the weaknesses of the study

6. PLOS authors have the option to publish the peer review history of their article (what does this mean?). If published, this will include your full peer review and any attached files.

Reviewer #1: No

Reviewer #2: **Yes: **Miguel Angelo Goes

---

## [Author Response · Author response to Decision Letter 0]

5 Aug 2022

Dear Editor, 

We thank the reviewer for their valuable comments. We have now revised the manuscript and incorporated the suggested changes that helped improve our manuscript. Please find below a point-by-point rebuttal to the reviewers’ comments. 

5. Review Comments to the Author

Reviewer #1: The present paper explores different patient-level factors related to COVID-19 severity and fatality in a tertiary care centre in Lebanon.

Major comments

§ The abstract should be re-structured, accounting for the importance of creatinine on COVID-19 severity in the introduction, for example. Methods should reflect the statistical techniques used throughout the paper ( i.e., regression, correlation tests, etc.). The results lack statistical measures, including coefficients or p-values for the main predictors. The discussion should account for the step forward linking creatinine and COVID-19 severity and how policies could be oriented for better diagnosis or treatments, etc. Also, those correlations exhibited are positive? Negative? Etc. It’s worth expanding on the main results of the research in the abstract.

The abstract has been revised according to the comments of the reviewer. The methods now reflect the statistical tools used. The statistical measures (P values) are now included. We simply omitted the Coefficients in the abstract (all detailed in the results section) to keep it easier to read. The results are expanded and show the positive association. The discussion reflects the policy implications.

§ Could you please clarify the epidemiological, demographic, clinical and laboratory data retrieved in the methods section? This should be in order and right after you mention them all. Now you only mention clinical data. Also, the measurement might differ from study to study; would you mind adding more information on those characteristics and how they were measured? (It could be added in the supplementary material).

As suggested by the reviewer, we have now expanded the materials and methods section and we clarified the data used in the analyses. We included in the revised manuscript the measurements used in the study and all related information.

The following text was added:

“After weight determination and height measurements, the body mass index (BMI) was calculated, and patients were evaluated for the following clinical manifestations: Abdominal pain, diarrhea, anosmia, fever, chills, fatigue, myalgia, headaches, cough, sore throat, and shortness of breath. Patients’ medical history were assessed for the following medical conditions: Asthma, chronic obstructive pulmonary disease (COPD), cancer, chronic kidney disease, cardiovascular disease (CVD), diabetes, hypertension, and autoimmune disease. Oxygen requirement, the need for and the duration of intubation, and the length of hospital stay until discharge or death were recorded. 

White blood and platelets counts were obtained on hospital admission. Liver function tests (serum glutamic-oxaloacetic transaminase (SGOT) or aspartate aminotransferase (AST), and serum glutamic pyruvic transaminase (SGPT) or alanine aminotransferase (ALT)); high-sensitivity C-reactive protein (CRP); hemoglobin (Hg); Serum Creatinine kinase; D dimer and ferritin levels were also determined on admission. The two main outcome variables tested were fatality or “severe” COVID-19 (defined as requiring intubation).”

§ Could you please clarify the main outcomes and how they are measured in the methods section? The statistical analysis section is only for statistical inference approaches, and therefore, the “Categorical and quantitative variable definitions are defined in Table S1. The two main outcome variables tested were fatality or “severe” COVID-19…..” should be moved up to patients characteristics or data collection. 

We moved all text related to patients’ information into the appropriate section as suggested. 

§ Statistical analyses. You speak about correlation in the abstract, but you used a logistic regression model, as this section depicts. Try to be clear on the regression method used. Did you test your outcomes? (i.e., their distributions, model residuals, multicollinearity, etc.). Also, did you use robust standard errors and if not, why? 

The word “correlation” was replaced with “association” in the Abstract, and throughout the manuscript.

Robust standard errors and other tests involving homoscedasticity in OLS were not relevant to the tests we applied. The logistic regression model was implemented using a glm predicting binomial outcomes, so residuals are not appropriate.

§ Logistic model using severe COVID-19 as the outcome variable. Did you compare severe COVID-19 against those patients having mild COVID-19, asymptomatic COVID-19 and those who did not have it? I think this has to be clarified, and comparison groups should be ordered more clearly. 

All enrolled patients were Covid-19 positive. We identified those requiring intubation as “severe.” The remaining were not severe. The statement defining “severe” was moved to the “Patients” section:

“The two main outcome variables tested were fatality or “severe” COVID-19 (defined as requiring intubation).”

§ You applied a Cox regression for the mortality outcome. Could you please mention it in the statistical analysis subsection? Also, you might explain a bit about the primary predictors/exposures, model structure, how time was handled, censoring, etc.

The statistical analysis section has been modified to read:

“Data were managed using pandas (version 1.1.2)(16). Multivariate logistic regressions were performed using statsmodels (version 0.12.0) (17) and Cox regressions were performed using coxph in R (survival ver 3.2.1, survminer version 0.4.9) (18) . Times were defined in terms of admission and discharge/fatality for COVID-19, with no other exclusions. Only records that completed discharge or fatality were included in all analyses. There were no outpatient follow-ups. Categorical and quantitative variable definitions are presented in Table S1. Summaries of the quantitative/categorical data were generated using pandas.”

§ The results section should be restructured. The explanation of variables incorporated in the modelling should be added in the methods section to facilitate readers’ understanding and information flow. 

As suggested by the reviewer, we have now re-structured the results section and we explained the variables incorporated in the modelling in the materials and methods section.

§ Supplementary material Table S1 should contain more detailed information. Were the variables analysed continuous or categorical? Why were the variables re-scaled, and does that impact the analyses? If they were re-scaled and centred, how and why did you do that?

For logistic regressions, category assignments generally were reduced to categorical variates according to clinically specified thresholds. The following has been added to the manuscript:

“Continuous variables including platelet counts, oxygen volume required, CRP, and Hb were scaled and centered by mean and standard deviation. Thresholds for assigning categories derived from continuous variables are included in Supplementary Table 1”

Supplementary Table 1 has been revised and shows thresholds and categories assignments based on continuous variables.

§ No limitations are mentioned in the discussion section, and attention should be drawn to population characteristics and sample size, lack of weighting techniques or representability of the whole population, etc.

We have now included a limitations section in the discussion, it reads as follows: 

“The absence of creatinine baseline levels (prior to admission) was a major limitation of the study. Moreover, markers of kidney function (other than creatinine) were not assessed upon admission. This was partly mitigated by the information collected on chronic kidney disease upon admission. The disease evolution could not be assessed against creatinine levels as additional data points on creatinine levels during hospitalisation were not available. In addition, follow-up data on patients after discharge was not available.”

§ What are the study's main implications, or how can we connect this information with the current situation in Lebanon where people are being vaccinated (40% at least with one dose, 35% with two doses, and 9% with booster)?

We have added the following paragraph to the discussion to reflect the main finding of the study in a vulnerable not yet vaccinated population: 

“Since the number of fully vaccinated subjects in Lebanon has not yet reached the 50% mark, it is therefore imperative to actively target patients with acute or chronic kidney injury for a full vaccination regimen to avoid COVID-19 associated morbidities and mortalities.” 

§ Main results tables were not attached in the PDF file. 

We will make sure when we resubmit to check for this error

Minor comments:

§ Please add the line numbers for the next revisions and to facilitate the review 

The line numbers are now added to the revised manuscript.

§ Please consider revising the main text; there are some spelling, format and text errors. For instance, “..In January 2020, The..” : the “the” shouldn’t be written in capital letters.

The text was revised and corrected.

§ Spell out WHO once first mentioned

We spelled out WHO (World Health Organization)

§ The term ‘vs’ should be written in formal English.

We changed “vs” to “versus” throughout the manuscript. 

§ Avoid the following words: “at the time of this writing”. Try to use the exact date to refer to the time and only if it is strictly needed.

As suggested by the reviewer, we removed “at the time of this writing” and only kept the date if needed.

§ Would you mind referring to the hospital analysed on the attended population, whether it is representative of the whole Lebanese population, etc. 

We modified the text, it now reads as follows:

“We investigated 819 COVID-19 patients who were admitted between January 2020 and April 2021 to the COVID-19 ward at a major tertiary care hospital in Lebanon. This hospital was one the major hospitals in the Lebanese capital that was dedicated to treating COVID-19 patients who came not only from the greater Beirut area but from across the entire country.”

§ If you first use WHO, then do not use its full name throughout the text. 

We refer to the World Health Organization as WHO after we spelled it out. 

§ Do not repeat over and over that 819 patients were admitted because it is mentioned in the methods and should be shown in descriptive stats tables.

As suggested by the reviewer, we deleted the repetitive reference to the 819 patients. 

§ Please spell out the M and F in the passage “66.77% M and 33.33% F”.

We spelled out Male (M) and female (F). 

§ There are some inconsistencies with the use of decimal figures.

We revised the decimal figures and made them consistent.

§ For p=0.000 is better to use p<0.001.

We changed p=0.000 to p<0.001 across the revised manuscript and tables.

§ There is no need to repeat ethical considerations two times throughout the manuscript.

We deleted the repetitive reference to Ethical Considerations. 

§ Spell out the acronyms once first mentioned

We spelled out all acronyms once first mentioned 

§ Define that COVID-19 intensity is defined by whether patients were intubated. 

We describe now very clearly that COVID-19 severity: means necessitating intubation. 

§ Consider reframing the title by adding that it is based in a tertiary-care hospital in Lebanon. 

We modified the title (based on reviewer 2 and editorial comment) and we made explicit reference to the study site and population throughout multiple sections of the manuscript. 

§ Consider proofreading your manuscript.

We revised the manuscript and corrected it. 

Reviewer #2: Dear Authors The current case report is interesting. But I have some opinions and doubts.

- Title Creatinine levels on admission are main modulators of COVID-19 severity.In my opinion, it would be better to inform there is an association between severe COVID-19 and serum creatinine levels or kidney function. 

The title was changed as recommended. Now it reads: 

“Creatinine levels on admission are associated with COVID-19 severity”.

- In abstractResultsCorrelation analysis of various comorbidities revealed that hypertension, diabetes,being overweight, kidney disease,.../What kind of kidney disease? Would it be CKD? AKI? Nephritis? Or AKD?

Information on kidney disease upon admission was CKD. Elsewhere in the manuscript, high creatinine level was used as a surrogate to define acute kidney disease. These were adjusted throughout the manuscript and tables when necessary.

- MethodsThis study investigated 819 COVID-19 patients admitted to the COVID-19 ward at atertiary care center in Lebanon and evaluated their vital signs and biomarkers./ Are the biomarkers from hospital admission? 

All vital signs and biomarkers were performed at the hospital laboratory upon admission. This is now reflected in the revised version of the manuscript.

- Materials and methodsPatientsClinical data on admission included clinical symptoms, biochemical markers on admission, previous medical conditions.../You should report what symptoms were studied, and what kinds of biochemical markers were used to compute.What were the medical conditions you registered?And about outcomes, did you compute the frequency of COVID-19-related AKI? And the need for kidney replacement therapy? 

The materials and methods section has been expanded. It now clearly indicates what medical conditions are registered and what outcomes are correlated with these. 

-ResultsIn my opinion, you should put the results of the multivariate analysis in a table-Common comorbidities for intubation and fatalityWhat was the type of kidney disease? Was it CKD? or glomerulonephritis with therapy immunosuppression?We interrogated hypertension, diabetes, being overweight (BMI ≥ 25kg/m2), kidney disease, CVD...What was the CI you used? Was it 95%?Overall, the intubation of COVID-19 subjects was highly associated with fatality (OR 71.20, CI[41.33-122.64], p <0.001) and with the oxygen consumption volume (OR, 3.18 CI[0.02-0.06], p<0.001)… so forth-Biomarkers of intubation and COVID-19 fatalityAgain, what was the CI you used? Was it 95%?We investigated serum levels of platelets, ferritin, CRP, Hb, creatinine, WBC, SGOT and SGPT and analyzed these using intubation as an independent variable. CRP (OR 1.48, CI [1.21-1.81], p=0.000), WBC (OR 2.70, CI [1.68-4.32], p=0.000), SGOT (OR 3.08, CI [1.88-5.03], p=0.000)./I suggest to write p<0.001 rather than p=0.000. …so forth. 

All occurrences of “CI” have been replaced by “95%CI.”

-Discussion In my opinion, you should report on the weaknesses of the study

We added a limitations section to the discussion

Additional Editor Comments:

The paper and results are interesting, but the paper suffers from many severe flaws.

- Please change the title. This is a correlational analysis of ~800 patients not an RCT, thus the claim in the title is not correct and misleading.

The title has been changed.

- The writing throughout is poor and needs revision, please have it proofread more carefully.

The manuscript has been carefully revised. 

- Additionally, the structure of the paper should adhere to scientific writing standards, do not put methods in the results, for example, clearly explain your cohort, etc, and put the tables in the paper not as some additional download.

The structure of the paper has been modified to adhere to scientific writing standards.

- On the methods, there needs to be significantly more included on the data, including, for example, what symptoms were extracted and when during their stay. 

The methods section has been expanded. Please see replies to the reviewers above. 

- The statistical analysis is also lacking in detail. Please refer to other papers published for advice on how to expand this section, but for example, "Multivariate logistic regressions were performed using statsmodels (ver 0.12.0) (17) and coxph in R (survival ver 3.2.1, survminer ver 0.4.9) (18)" tells the reader almost nothing about the methods and sticks in that a cox was performed as if it were a logistic regression.

Statistical analysis details have been added.

- Also, why would data be managed in pandas but stats run in R, that makes little sense, statsmodels has a survival function. 

Almost all the analysis was done using statsmodels. We chose the R package for Cox primarily due to prior experience, all other considerations aside.

- On the results and interpretation, just because something is correlated does not make it causative. Please remove any reference to creatinine levels being the causitive. The reviewers both have additional substantial concerns about the methods employed and you must add a robust limitations section.

The text has been revised and any reference to creatinine level is causative has been removed.

The statistical methods employed have been described in detail to address the reviewers' concerns.

---

## [Decision Letter · Decision Letter 1]

31 Aug 2022

PONE-D-22-11784R1Creatinine levels on admission are associated with COVID-19 severityPLOS ONE

Dear Dr. Zalloua,

Thank you for submitting your manuscript to PLOS ONE. After careful consideration, we feel that it has merit but does not fully meet PLOS ONE’s publication criteria as it currently stands. Therefore, we invite you to submit a revised version of the manuscript that addresses the points raised during the review process.

Thanks for your earlier revisions, they have improved the paper, but unfortunately, while the reviewers approved of your changes, the paper is still not meeting criteria for publication in PLOS One. The major issue as I detail below is that I can't follow the statistics. The methods describe multivariate regressions, but I don't really see the results - though this may just be a problem related to how the results are described. Which brings me to the second issue which is the presentation of the article. While the writing is improved, it still needs work (in line with publication criteria #5). It would greatly aid the paper if you used the STROBE guidelines as well to help ensure the study is fully described. Thanks for the efforts here and I look forward to your next revision.

We look forward to receiving your revised manuscript.

Kind regards,

Eili Y. Klein, PhD

Academic Editor

PLOS ONE

Additional Editor Comments:

While the reviewers were satisfied with the revisions, there are still some issues with the language in the paper that makes it confusing and hard to read. I have made many suggested edits in the revised draft. I would also suggest having a editor review the paper to ensure compliance with criteria for english presentation (see #5: https://journals.plos.org/plosone/s/criteria-for-publication). Additionally, I am unable to fully follow the results as it relates to the methods. The methods suggest that multivariate logistic regressions were done, yet I only see binomial regression results. This may just be mislabeled - but needs to be clarified in line with publication criteria #3. I would highly recommend that the authors follow the STROBE guidelines for reporting on observational studies (https://www.ncbi.nlm.nih.gov/pmc/articles/PMC2870880/), that will ensure that the results are clear. Finally, the title suggests just creatinine levels were important, but that is only part of the story as creatinine was highly associated with hypertension, and it probably is beyond the capability of this study to disentangle the relationship. I would suggest that the title more accurately reflect the study type (as per STROBE guidelines) as well as more accurately reflect results. For instance Observational study of factors associated with Morbidity and Mortality from COVID-19 in Lebanon, 2020-2021 or something similar. Overall I still feel this can be a publishable paper, but still needs some work to meet PLOS One publication criteria.

Minor comments

1. Add timeframe to methods

2. Remove use of correlation where not appropriate

3. Reduce the introduction length by removing extraneous passages and unlinked information

4. Please ensure the correct tense is used throughout

5. Be more clear about the other biomarkers and why only creatinine is your focus given that it was so highly correlated with hypertension

Reviewers' comments:

Reviewer's Responses to Questions

**Comments to the Author**

1. If the authors have adequately addressed your comments raised in a previous round of review and you feel that this manuscript is now acceptable for publication, you may indicate that here to bypass the “Comments to the Author” section, enter your conflict of interest statement in the “Confidential to Editor” section, and submit your "Accept" recommendation.

Reviewer #1: All comments have been addressed

Reviewer #2: All comments have been addressed

2. Is the manuscript technically sound, and do the data support the conclusions?

Reviewer #1: Yes

Reviewer #2: Yes

3. Has the statistical analysis been performed appropriately and rigorously? 

Reviewer #1: Yes

Reviewer #2: Yes

4. Have the authors made all data underlying the findings in their manuscript fully available?

Reviewer #1: No

Reviewer #2: Yes

5. Is the manuscript presented in an intelligible fashion and written in standard English?

Reviewer #1: Yes

Reviewer #2: Yes

6. Review Comments to the Author

Reviewer #1: Congratulations to the authors of the article. The English can certainly improve but its legible. All comments I made were correctly addressed. Therefore, I dont have more comments for the authors.

Reviewer #2: (No Response)

7. PLOS authors have the option to publish the peer review history of their article (what does this mean?). If published, this will include your full peer review and any attached files.

Reviewer #1: No

Reviewer #2: No

---

## [Author Response · Author response to Decision Letter 1]

2 Sep 2022

Editor’s comments:

The methods suggest that multivariate logistic regressions were done, yet I only see binomial regression results. This may just be mislabeled - but needs to be clarified in line with publication criteria #3. I would highly recommend that the authors follow the STROBE guidelines for reporting on observational studies (https://www.ncbi.nlm.nih.gov/pmc/articles/PMC2870880/), that will ensure that the results are clear. Finally, the title suggests just creatinine levels were important, but that is only part of the story as creatinine was highly associated with hypertension, and it probably is beyond the capability of this study to disentangle the relationship. I would suggest that the title more accurately reflect the study type (as per STROBE guidelines) as well as more accurately reflect results. For instance Observational study of factors associated with Morbidity and Mortality from COVID-19 in Lebanon, 2020-2021 or something similar. 

We thank the editor for pointing out this matter and we regret the confusion. 

We used the term multivariate logistic regressions since our linear models included multiple factors, such as adjustment variables, and we did not state that we performed multinomial tests. We realize that this terminology is confusing since our multivariate regression was in fact a simple binomial logistic regression (in the tables we used that term). We have now removed the term multivariate, and we used logistic regression throughput. We kept the label binomial in the Tables’ captions. We also followed the STROBE guidelines as suggested. We addressed all other comments raised in the revised version of the manuscript and these include a change of the title, and tables as suggested.

Minor comments

1. Add timeframe to methods

We added the timeframe in the methods section of the Abstract.

2. Remove use of correlation where not appropriate

We inadvertently missed the removal of correlate/correlation from the previous submission although we have indicated this change in the rebuttal. We thank the editor for pointing out to this input error. We have now removed correlate/correlation from the revised manuscript.

3. Reduce the introduction length by removing extraneous passages and unlinked information

We revised the introduction, and we removed extraneous passages and unlinked information.

4. Please ensure the correct tense is used throughout

We ensured that the correct tense is used throughout.

5. Be more clear about the other biomarkers and why only creatinine is your focus given that it was so highly correlated with hypertension

We clarified this aspect in the discussion, and we shed light on the interaction between creatinine and hypertension.

---

## [Editor Report · Decision Letter 2]

11 Sep 2022

Observational study of factors  associated with Morbidity and Mortality from COVID-19 in Lebanon, 2020-2021.

PONE-D-22-11784R2

Dear Dr. Zalloua,

We’re pleased to inform you that your manuscript has been judged scientifically suitable for publication and will be formally accepted for publication once it meets all outstanding technical requirements.

Kind regards,

Eili Y. Klein, PhD

Academic Editor

PLOS ONE
---

## [Editor Report · Acceptance letter]

10 Oct 2022

PONE-D-22-11784R2 

Observational study of factors associated with Morbidity and Mortality from COVID-19 in Lebanon, 2020-2021. 

Dear Dr. Zalloua:

I'm pleased to inform you that your manuscript has been deemed suitable for publication in PLOS ONE. Congratulations! Your manuscript is now with our production department. 

Kind regards, 

on behalf of

Dr. Eili Y. Klein 

Academic Editor

PLOS ONE